# Development of the Circular Economy Design Guidelines for the Australian Built Environment Sector

Atiq Zaman [1] , Ana Maria Caceres Ruiz [1,*], Salman Shooshtarian [2], Tim Ryley [3] , Savindi Caldera [3] and Tayyab Maqsood [2]

1    Curtin University Sustainability Policy Institute, School of Design and the Built Environment, Curtin University, Perth 6102, Australia
2    School of Property, Construction and Project Management, RMIT University, Melbourne 3001, Australia
3    Cities Research Institute, Griffith University, Brisbane 4111, Australia
*    Correspondence: anamaria.caceresruiz@curtin.edu.au; Tel.: +61-8-9266-9018

**Abstract:** The construction and demolition (C&D) waste stream is the main source of solid waste in Australia. While there is a strong circularity drive in Australia's and state/territory governments' waste regulatory framework, clear guidelines for C&D waste management are yet to be developed for the built environment sector in Australia. This study proposes a suite of construction industry-specific guidelines for achieving circular economy (CE) goals by reviewing issues related to "Design for Zero Waste" (DfZW) and "Design for Recycling" (DfR). To do so, this study explores the current CE practices in construction and infrastructure projects in both global and Australian contexts through a systematic literature review. In addition, barriers and enablers of CE in the built environment were identified. This study provides a list of guidelines that can help industry practitioners achieve CE in the construction sector in Australia. These guidelines draw on the main themes identified through the literature review: circularity practices, resource management, innovation, and optimisation. Thus, this study bridges the gap between theory and practice by offering clear, circular guidelines for designing out C&D waste in Australia. The proposed guidelines enable industry practitioners to keep products and materials in use for a longer period and develop strategies to regenerate natural systems. Future research should focus on several aspects, including measuring emissions reductions linked to the strategies shown in the proposed guidelines.

**Keywords:** circular economy; construction industry; built environment; guidelines; resource management; products with recycled content; design for zero waste; design for recycling

## 1. Introduction

The construction industry is both carbon-intensive and highly material and energy-demanding [1]. This sector contributes to a considerable portion of global energy-related greenhouse gas emissions (GHG) (over 35%) and consumes half of the global raw materials [1,2]. In addition, construction and demolition (C&D) waste generated during construction activities accounts for around one-third of global waste generation [3]. Australia has a higher C&D waste generation rate than the global average. The latest statistics have shown that Australians generated 29 million metric tonnes (Mt) of C&D waste in 2021–2022, which represents around 38% of Australia's entire solid waste generation [4]. Around 6.4 Mt of C&D waste was disposed of in landfill sites across the country [4], a considerable loss of valuable materials that could have been repurposed. Therefore, policy development needs to be further expanded to include new/current waste management schemes, including manufacturers' shared responsibility for waste generation, subsidies for C&D waste recycled materials, and the proximity principle [5].

A circular economy (CE) offers a framework to address the adverse effects of the business-as-usual model of the construction industry. A CE can be defined as an economic

system that replaces the end-of-life (EoL) concept by reducing, reusing, recycling, and recovering materials [6]. The redefinition of the EoL concept in a CE is a key factor in decoupling economic growth and resource consumption [7]. A CE has the overarching aim to achieve sustainable development [6] as it works towards creating social, economic, and environmental value [8]. The Ellen MacArthur Foundation (EMF) has established three main principles of a CE: (i) eliminate waste and pollution, (ii) circulate products and materials, and (iii) regenerate nature [9]. A CE in the built environment aims to use sustainable materials, to keep the value of materials, to increase material recovery once these have reached their EoL, and to minimise waste [10,11]. This coincides with the (i) and (ii) EMF CE principles. This study agrees with this circular built environment definition and considers that restoring nature is also a key part of a CE in the built environment.

Several countries and regions around the world have adopted a CE through action plans to strive for a cleaner and more competitive future. The European Green Deal, Europe's new agenda for sustainable development, has established the 2020 CE action plan as a key cornerstone [12]. One of the measures of the action plan includes focusing on the most resource-intensive sectors where there is more potential for circularity, including construction and buildings [12]. A similar trend was identified in other countries. In 2021, China released a CE multi-year plan to maximise resource use and productivity through improved recycling systems and increased use of renewable sources [13]. In India, a multi-stakeholder consultation took place in 2022, where the development of a framework to support and implement CE models across sectors was discussed [14].

Through CE practices embedded with 'design for zero waste' (DfZw) and 'design for reuse/recycling' principles, much of the C&D waste generation could have been avoided or at least recirculated within the construction sector supply chain. In fact, evidence shows that there may be a positive correlation between new construction, waste recovery, and energy consumption. Although there is no clear academic consensus on the environmental benefits of using recycled materials, some studies have shown, through life-cycle analysis of either costs or energy, that the use of products with recycled content (PwRC) and pre-fabricated components in new construction has several positive impacts. Some benefits include GHG emissions reduction, non-renewable energy savings, and less water consumption and hazardous waste generation [15–17]. This is key to addressing one of the outcomes of the recent COP27, which is the need for a transition in patterns of consumption and production through several measures, including transitions to renewable energy [18,19]. Additionally, other indirect emissions from the sector may have been decreased in addition to those from C&D waste.

On average, 10% of business-related GHG emissions are classed as scope 1 and scope 2, while 90% are associated with scope 3 emissions [20]. The three scopes classification is a way to categorise the various types of emissions that a business creates during its operation and across its entire value chain [21]. The term "scope" was assigned by the Greenhouse Gas Protocol, which has developed internationally accepted standardised frameworks to enable the management and measurement of GHG from businesses' value chains [21,22]. Scope 1 emissions come from business activities (e.g., emissions from the company's facilities and/or from the company's vehicles and equipment), and Scope 2 is from purchased energy for business use (e.g., heating, cooling, electricity) [23]. Scope 3 emissions are indirect industry emissions that are generated during upstream business activities (e.g., capital goods, business travel, transportation and distribution of the product between tier 1 supplier and company, waste produced from upstream phases), as well as downstream business activities (transportation and distribution of products between company and end consumer, use of products, waste produced from downstream operations, end-of-life management of products) [23]. Figure 1 shows the main sources of scope 1, 2, and 3 emissions. Given that one of the principles of a CE is designing out waste pollution and emissions generated pollution, circular solutions along the entire built environment industry value chain will lower scope 1, 2, and 3 emissions [20].

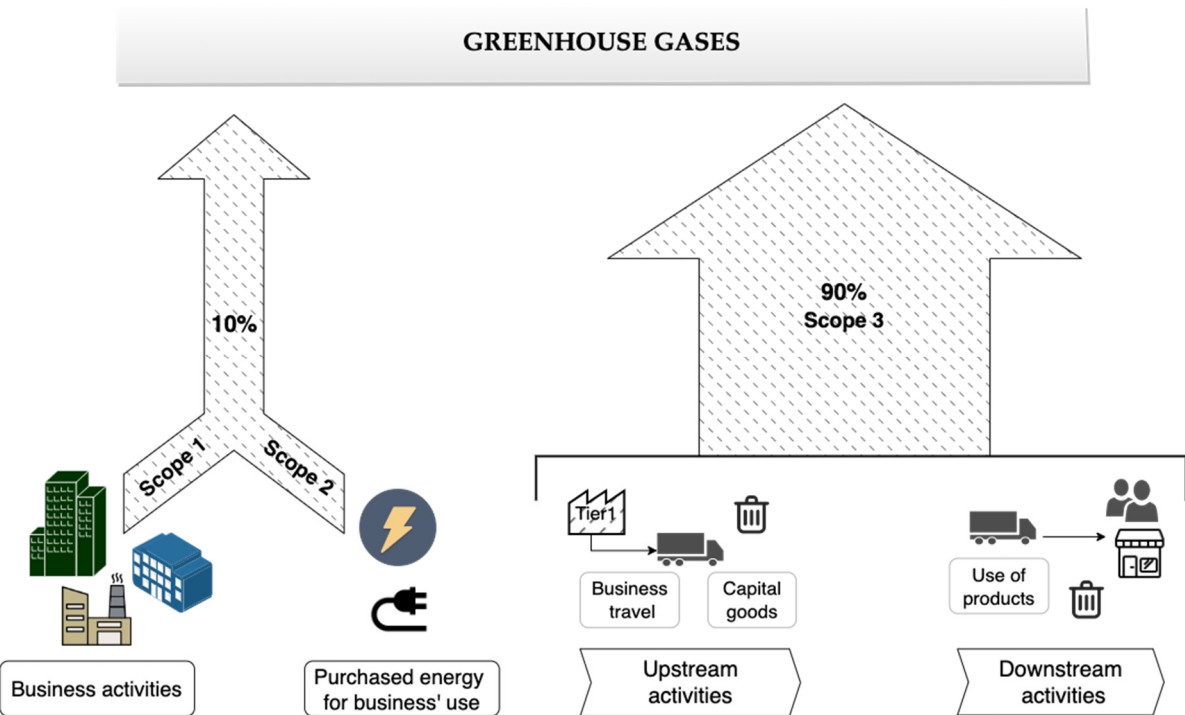

**Figure 1.** Main sources of scope 1, 2, and 3 emissions. Source: Authors.

Since the China Waste Ban, major changes in Australian regulatory policies have taken place to strengthen local waste infrastructure and to improve overall waste management practices based on sustainability principles [5,24]. Policies include the 2018 National Waste Policy and the 2025 National Packaging Targets. The National Waste Policy Action Plan (NWPAP) 2019 sets 7 targets, 17 strategies, and 78 actions to implement the 2018 National Waste Policy and to complement and support the implementation of the national packaging targets. The following targets are highly relevant for C&D waste management:

- Reduce total waste generated in Australia by 10% per person by 2030;
- An 80% average resource recovery rate from all waste streams following the waste hierarchy by 2030;
- Significantly increase the use of PwRC by governments and industry by 2030;
- Make comprehensive, economy-wide, and timely data publicly available to support better consumer, investment, and policy decisions.

### 1.1. Circular Economy Policies in States and Territories in Australia

The current National Waste Policy (2018) was prepared by the federal, state, and territory governments, and it explicitly endorsed the transition into a CE. The 2018 National Waste Policy, which came into effect with the 2019 National Waste Policy Action Plan, sets a framework for waste and recycling in Australia [25]. Following the National Waste Policy, each state and territory in Australia has outlined its waste strategy with relevant targets and priorities [26].

Relevant to this research and in line with the Commonwealth Government, states and territories in Australia have incorporated or have started to consider the CE in their waste management strategies. The application and practices of a CE in the state and territory governments vary depending on their local context and priorities. For example, South Australia (SA) has a long history of driving sustainable reuse of waste through policy and legislation spanning more than 30 years [1] (e.g., the Container Deposit Scheme). Moreover, the SA Government has also initiatives to develop partnerships to implement the CE in regional areas [27]. New South Wales (NSW)'s recent CE policies focus on eight key areas which emphasise the importance of 'push' and 'pull' initiatives to stimulate both the

supply and demand sides of resources in the economy. The transition to a CE is enabled by data, investment, innovation, collaboration, skills, and engagement [28].

'Recycling Victoria' is the Victorian Government's 10-year policy and action plan for waste and recycling with 26 action items to achieve four CE goals [27]:

- Goal 1—Design to last, repair, and recycle;
- Goal 2—Use products to create more value;
- Goal 3—Recycle more resources;
- Goal 4—Reduce harm from waste and pollution.

The Australian Capital Territory (ACT) Government aims to achieve a CE by considering five main factors: civil works (reduce emissions), infrastructure (increased use of PwRC), building materials (encourage the use of recycled building materials), and residents and business (support them to reduce consumption). The ACT Government's CE strategies are highly relevant for C&D waste as it specifically considers emissions reduction and increased use of PwRC in civil and infrastructure works and building construction projects.

The CE has been incorporated into Western Australia (WA)'s Waste Avoidance and Resource Recovery Strategy 2030 and into the Tasmanian Government's draft Waste Action Plan. The Northern Territory (NT) Government has recently published its waste strategy, where the government will work in partnership with local governments, industry, non-government organisations, and communities toward the following three strategic but ambitious targets:

- Priority 1: Modernise the regulatory framework to protect the environment and create the right regulatory settings for growing the CE-driven industry;
- Priority 2: Start transitioning the Territory to a CE;
- Priority 3: Establish the CE-driven industry as a contributor to the territory's AUD 40 billion by 2030 vision.

Table 1 provides a summary of waste strategies policies for each state, identifying the C&D targets per policy.

**Table 1.** Summary of the States and Territory Governments' CE-related policies in Australia.

| States and Territories | Waste/CE Strategy | Action Plan/Priority Areas/Targets | Specific to C&D Waste | C&D Waste Targets |
|---|---|---|---|---|
| ACT [29] | ACT Waste Management Strategy | Carbon neutral waste sector | No | No |
| NSW [30] | NSW Circular Economy Policy Statement- Too Good to Waste | Eight priority areas and three recycling targets | Yes | 80% for construction and demolition waste |
| NT [31] | The Northern Territory Circular Economy Strategy 2022–2027 | Three priority areas | No | No C&D-related targets |
| QLD [32] | Waste Management and Resource Recovery Strategy | Targets for 2050<br>- 25% reduction in household waste<br>- 90% of waste is recovered and does not go to landfill<br>- 75% recycling rates across all waste types | Yes | 85% of C&D waste diversion to landfill by 2050 |
| SA [33] | South Australia's Waste Strategy 2020–2025 | By 2025:<br>- 95% diversion of C&D | Yes | 95% diversion of C&D by 2025 |
| TAS [34] | Draft Waste Action Plan, 2019 | - 10% waste reduction by 2030<br>- 80% resource recovery by 2030 | Generic | No |

**Table 1.** *Cont.*

| States and Territories | Waste/CE Strategy | Action Plan/Priority Areas/Targets | Specific to C&D Waste | C&D Waste Targets |
|---|---|---|---|---|
| VIC [35] | Recycling Victoria A new economy | • Divert 80% of waste from landfills by 2030 <br> • Cut total waste generation by 15% per capita by 2030 | Generic | No |
| WA [36] | Waste Avoidance and Resource Recovery Strategy 2030 | • 30% reduction in C&D waste generation per capita by 2030 <br> • Increase C&D material recovery to 80% <br> • 85% waste diversion rate | Yes | 80% material recovery target by 2030 |

### 1.2. Research Gap, Aim, and Objectives

Despite having a strong emphasis on CE principles in the national and state/territory governments' policies [37], clear guidelines for C&D waste management are yet to be developed for the built environment in Australia. From the literature, there are various design approaches to minimise C&D waste, namely, Design out Waste (DoW) [38]. This practice promotes the most efficient use of available materials to reduce the number of materials ultimately used in the construction processes [38]. Nevertheless, DoW guidelines for achieving the best waste management plan have not been thoroughly outlined yet [38]. Therefore, this study aims to review the issues related to "DfZW" and "Design for Recycling" (DfR) to help develop industry-specific CE guidelines. The study is part of a larger national study (Project 1.85—Enhancing the use of products with recycled contents in the Australian construction industry) that aimed to guide government decision-makers and industry practitioners to facilitate the utilisation of PwRC and to contribute to the further development of the CE in the built environment sector.

The following objectives are considered to achieve the aim of the study:

1. To examine the current CE practices in the built environment in the global and Australian context;
2. To identify the key barriers and enablers of applying CE principles in the built environment sector;
3. To develop industry-specific guidelines for implementing CE practices in the construction and infrastructure industry.

## 2. Methodology

### 2.1. Research Strategy

This study employs a structured qualitative content analysis to collect data on the issues related to "DfZW" and "DfR" in the construction industry. This approach was inspired by "Preferred Reporting Items for Systematic Reviews and Meta-Analyses: the PRISMA) statement" by Moher, Liberati [39] and by the five key phases and eight steps outlined in "Producing a Systematic Review" by Denyer and Tranfield [40]. Scopus and Web of Science (WoS) were identified to be most appropriate for sustainability research and the topic of this paper. The search was carried out without year intervals. Peer-reviewed and open-access records were considered for this search. The subject area/category of environmental science was selected as the main field of this study. Grey literature was also included in the search, to make sure no relevant data from government and industry reports were overlooked. Figure 2 shows the research framework of the study.

### 2.2. Scope of Study and Search Strings Used

The study applied three-level (primary, secondary, and tertiary) search string criteria to identify relevant articles from more generic to specific topic areas. Table 2 shows the search strings used in different search criteria. At the end of 5 search attempts, 313 articles were preliminarily identified from both databases (133 from Scopus and 180 from WoS). Table 2 shows five search strings used, all based on the research framework. For instance, search attempt 1 was inspired by Design for Zero Waste (DfZw), as seen in Figure 2.

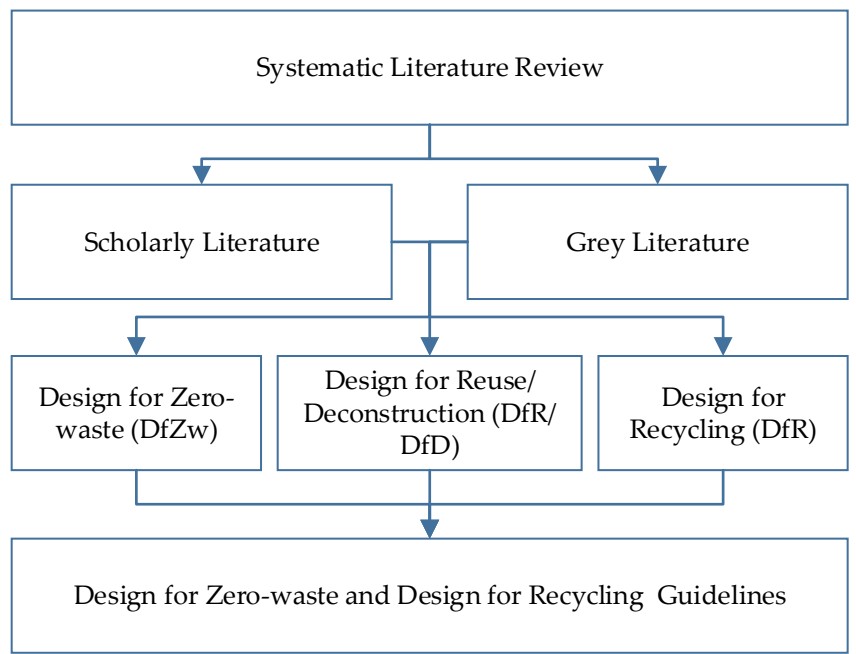

**Figure 2.** The research framework of the study.

**Table 2.** Search strings used in the PRISMA approach.

| Search Attempt | Primary Search Criteria | Secondary Search Criteria | Tertiary Search Criteria | No of Available Articles in the Primary Search | | No of Available Articles in the Secondary Search | | No of Available Articles in the Tertiary Search | |
|---|---|---|---|---|---|---|---|---|---|
| | | | | Scopus | WoS | Scopus | WoS | Scopus | WoS |
| 1 | Design for 'zero-waste' OR design for 'no waste' | Design for 'zero-waste' OR design for 'no waste' AND construction OR infrastructure project | (Design for 'zero-waste' OR design for 'no waste') AND (construction OR infrastructure project) AND (guidelines OR strategies OR policies OR procedures OR standards OR protocols OR framework) | 4228 | 384 | 89 | 20 | 43 | 38 |
| 2 | Design for reuse OR design for recovery | Design for reuse OR design for recovery OR design for deconstruction AND construction OR infrastructure project | (Design for reuse OR design for recovery OR design for deconstruction) AND (construction OR infrastructure project) AND (guidelines OR strategies OR policies OR procedures OR standards OR protocols OR framework) | 73,361 | 48 | 34 | 4 | 20 | 29 |
| 3 | Design for recycling OR design for resource recovery | Design for recycling OR design for resource recovery AND construction OR infrastructure project | (Design for recycling OR design for resource recovery) AND (construction OR infrastructure project) AND (guidelines OR strategies OR policies OR procedures OR standards OR protocols OR framework) | 3726 | 190 | 80 | 7 | 44 | 39 |
| 4 | Design for 'zero-waste' OR design for 'no waste' OR design for circular economy OR circular design | Design for 'zero-waste' OR design for 'no waste' OR design for circular economy OR circular design AND construction OR infrastructure project | (Design for 'zero-waste' OR design for 'no waste' OR design for circular economy OR circular design) AND (construction OR infrastructure project) AND (guidelines OR strategies OR policies OR procedures OR standards OR protocols OR framework) | 2324 | 603 | 12 | 35 | 7 | 48 |
| 5 | Design for deconstruction OR design for disassembly | Design for deconstruction OR design for disassembly AND construction OR infrastructure project | (Design for deconstruction OR design for disassembly) AND (construction OR infrastructure project) AND (guidelines OR strategies OR policies OR procedures OR standards OR protocols OR framework) | 25,599 | 332 | 569 | 35 | 19 | 26 |
| Total Identified articles from SCOPUS and WoS databases in different search levels | | | | 109,238 | 1557 | 784 | 101 | 133 | 180 |
| Total articles excluded due to eligibility check (duplication, non-relevant, etc.) | | | | 198 | | | | | |
| The selected articles for review (N) | | | | 115 | | | | | |

### 2.3. Data Synthesis

The following selection criteria were adopted to select studies with the most relevance to the objectives:

(1) Only journal articles written in English were considered in the search criteria;
(2) Only sources that have a primary focus on design were captured;
(3) For the grey literature, available reports, white papers, and working papers from the relevant governments (presented in Section 1) and industry bodies (e.g., GBA, AIC, AIA, AIB, etc.) were considered.

The articles found on both databases that met these criteria were exported for further analysis. Next, duplicated records were cleaned up. After this, the abstract, title, and

keywords were carefully assessed. Those articles deemed irrelevant were excluded. For instance, some papers were focused on health risks from construction materials, marketing, or wastewater. Figure 3 gives an overview of the articles' screening and selection process.

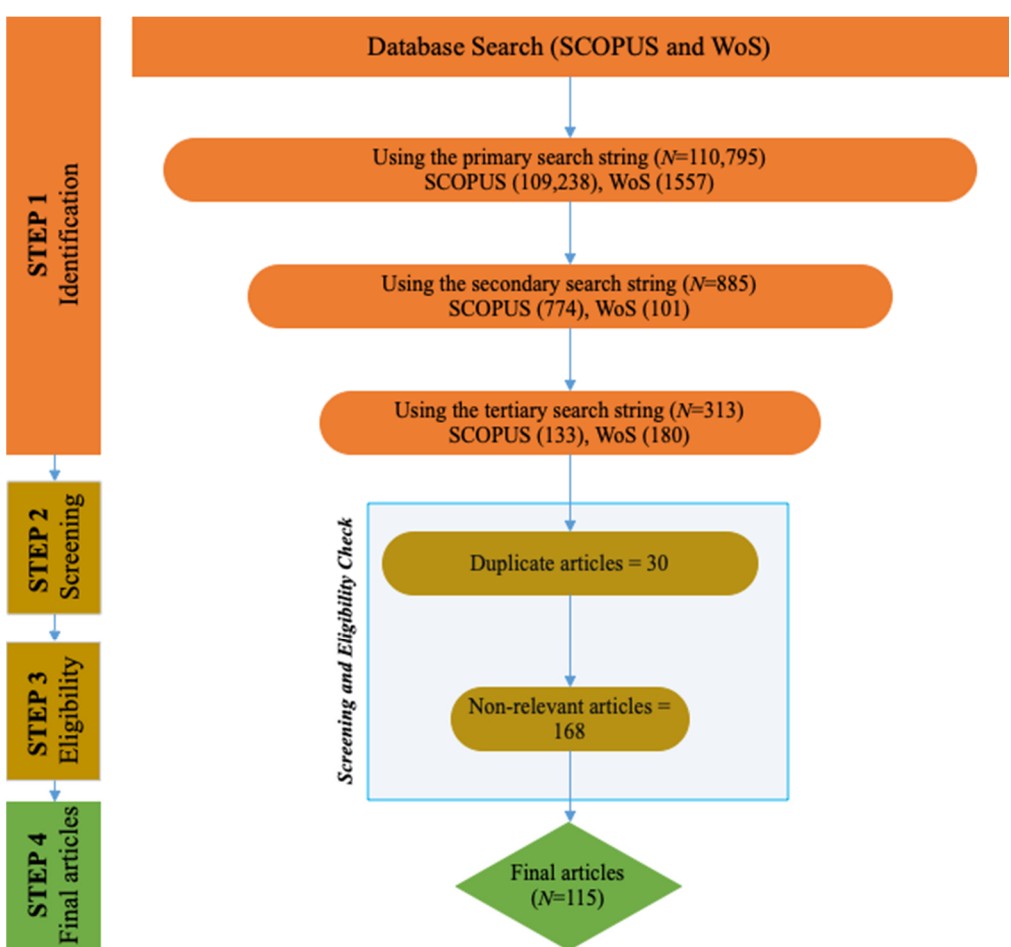

**Figure 3.** PRISMA flow diagram of the literature search process of articles' selection. Source: adapted from Moher, Liberati [39].

## 3. Results

### 3.1. Trends of the CE Studies

One finding from the selected articles is that Europe and Asia are leaders in CE research in the built environment (Table 3). Other regions in the world have done their part as well, although the number of articles is far fewer than the leaders.

The analysis of metrics pertaining to the selected articles metrics shows that there has been a considerable increase in the number of publications around the topic areas in the last 10 years (Figure 4). Around 70% of the articles (81) were published in 2010 and 2022, which shows an influx of research publications recently. In 2022 alone, the number of selected publications was about 2.5 higher than the year before.

The majority of the articles (67%) were considered to be on a building construction sector topic, whilst only 6% were deemed to be infrastructure related. The remaining 27% covered both project types. Around 63% of articles were published in traditional journals, and 37% were published in open-access journals. Based on the title, abstract, and keywords review of all 115 articles, the articles are grouped into 8 specific focus areas after completing multiple iterations of identification, selection, and regrouping of research focus and scope covered in the articles (Table 4).

**Table 3.** A summary of regions and countries of the selected studies.

| Region | Countries | No of Articles |
|---|---|---|
| Africa | Egypt and Nigeria | 9 |
| Americas | Brazil, Canada, and the USA | 10 |
| Asia | Indonesia, China, Hong Kong, India, Kazakhstan, Sri Lanka, Korea, Malaysia, Myanmar, Pakistan, Philippines, and South Korea. | 35 |
| Europe | Austria, Belgium, Finland, France, Italy, Germany, Poland, Luxembourg, Netherlands, UK, and Ukraine | 46 |
| Middle East | Iran, Oman, Saudi Arabia, UAE | 3 |
| Oceania | Australia | 11 |
| Global | Global | 1 |

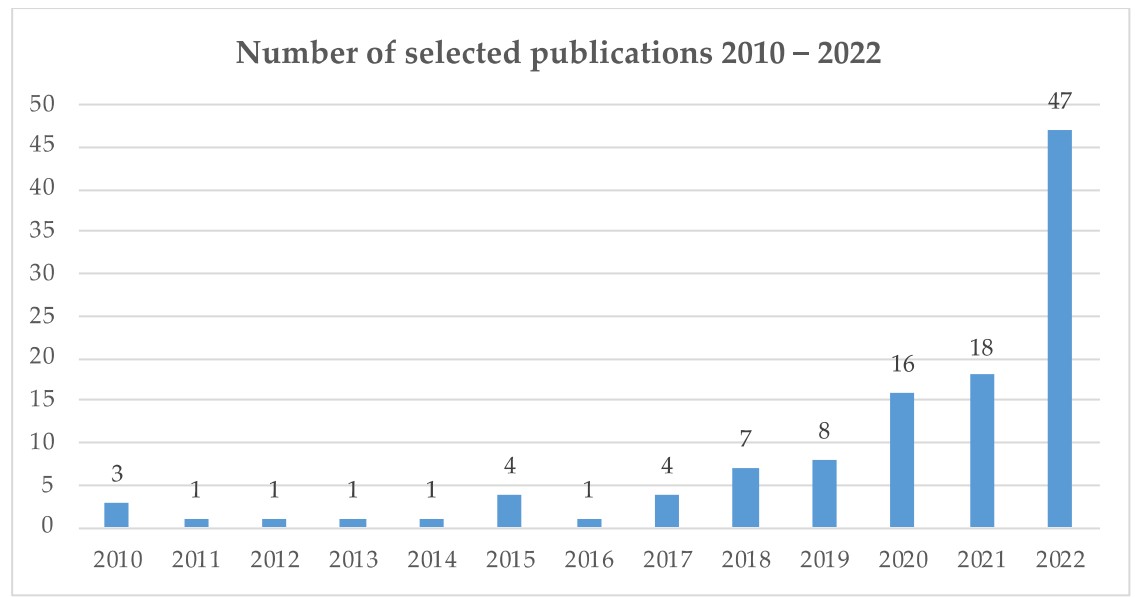

**Figure 4.** Selected articles for every year between 2010 and 2022.

**Table 4.** Broader scope, key specific focus, and elements covered in the article.

| Broader Scope | Specific Focus | Number of Articles | Key Elements Covered in the Articles |
|---|---|---|---|
| Circularity practices | Circular economy/zero waste/no-waste | 19 | Zero waste and CE principles, understanding of key barriers and opportunities in the transition to a CE-built environment. |
| | Design for deconstruction/disassembly (DfD) | 12 | The application of CE and zero waste design principles such as DfD or deconstruction. Design for reusability and adaptability and the opportunities for reusing the entire building by relocating and adaptive reuse. |
| | Reuse of building, elements, materials | 11 | The opportunities for reusing construction materials such as aggregates and timber, and reusing building elements/components such as steel frames and steel beams/columns. |

**Table 4.** *Cont.*

| Broader Scope | Specific Focus | Number of Articles | Key Elements Covered in the Articles |
|---|---|---|---|
| Resource management | Recycling/Resource management | 21 | Waste resource management from the point of minimisation, recycling, and recovery of resources from waste, including post-disaster events and the relevant regulatory policies. |
| Innovation and optimisation | Digital technology | 14 | Digitisation and application of innovative technologies such as BIM, digital material bank, material passport, and cloud computing to enable data-driven decisions are the key focus of digital innovation. |
| | Construction innovation | 13 | The recent innovation in the construction technologies such as off-site/modular construction, prefabrication, and green construction materials using alternative building materials sourced from recycled components seem to be the key areas covered by the articles. |
| | Project/supply chain optimisation | 11 | The project optimisation through innovative approaches such as the integration of a client–designer interface and introducing an early contractor's involvement in the design phase to minimise waste, better manage materials, and optimise productivity. |
| Sustainability Assessment | Measurement | 14 | Measuring impacts of current practices and the sustainability benefits and performance in the context of sustainability priorities. |

Figure 5a shows that research on current CE practices in the built environment has had a dominant theoretical focus rather than a practical one. Around two-thirds of the selected articles applied theoretical methodologies (literature reviews), while case studies and prototypes accounted for 22% and 13%, respectively. Figure 5b illustrates that technology, innovation, and environment are the most studied areas in the selected articles (between 22% and 23%). Less focus is given to economic (13%), policy (11%), and social (9%) aspects. Figure 5c gives an overview of the life cycle stage that the selected articles cover. The EoL stage makes up 42% of the research, with 22% addressing the EoL through processes such as disassembly and/or reuse, and 20% of the articles focus on the EoL from the waste management perspective. Nearly a quarter of the articles address the design stage, which is followed by both the construction and manufacture stages, at 18% and 15%, respectively. Interestingly, only 1% of the articles discuss the operation stage, showing a gap in research.

Appendix A shows the meta-analysis carried out on the selected articles. The categories analysed are the types of research methods applied, the sustainability dimension(s), technology, innovation, policy aspects, and the life cycle phase(s) addressed in the articles. A rating scale is used to show the extent to which each of these categories is studied in the articles. Progress bar icons are applied for this purpose.

*3.2. Key Barriers*

The application of the meta-analysis to the collected data served to identify the key barriers to implementing the CE in the built environment sector in Australia (Figure 6). The results suggested four key clusters of barriers: cultural, regulatory, organisational, and knowledge and understanding. The literature examined in this section covers both Australian and other countries' sources.

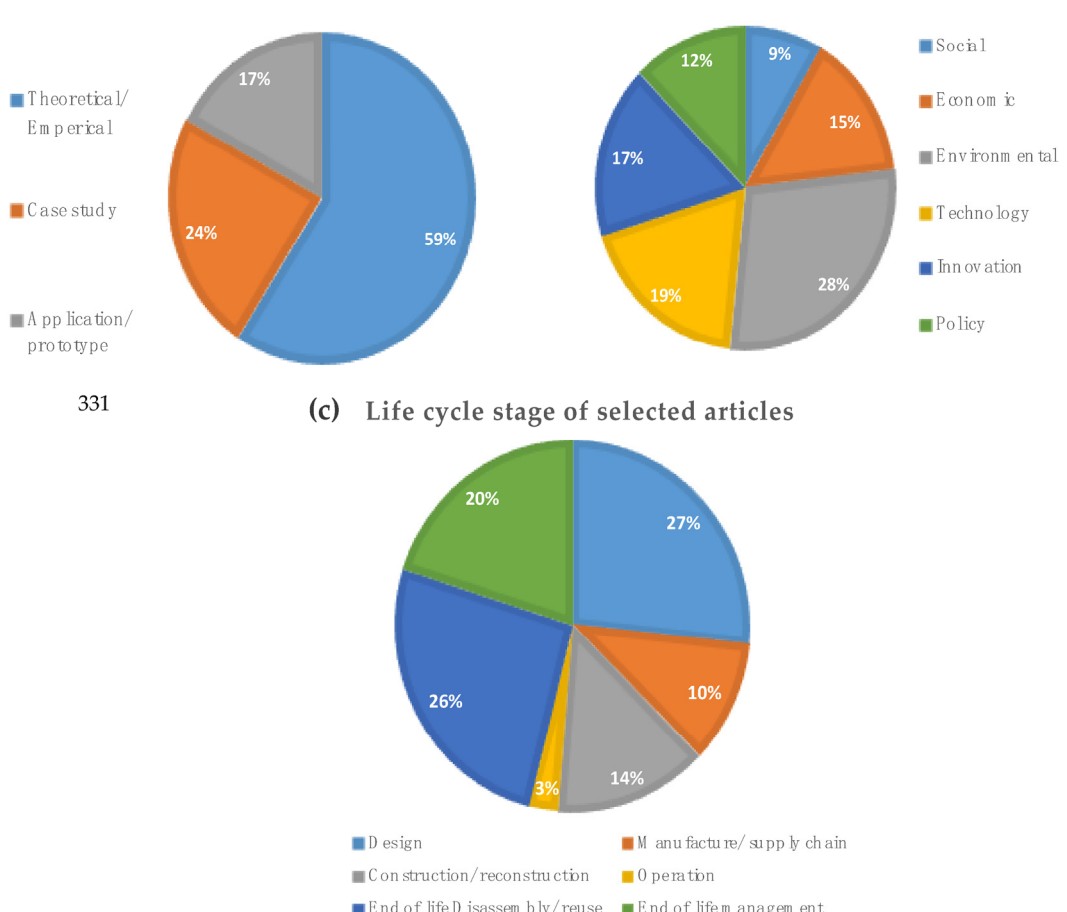

**Figure 5.** Statistics on the study's literature (**a**) research method, (**b**) focus area, and (**c**) life cycle stage.

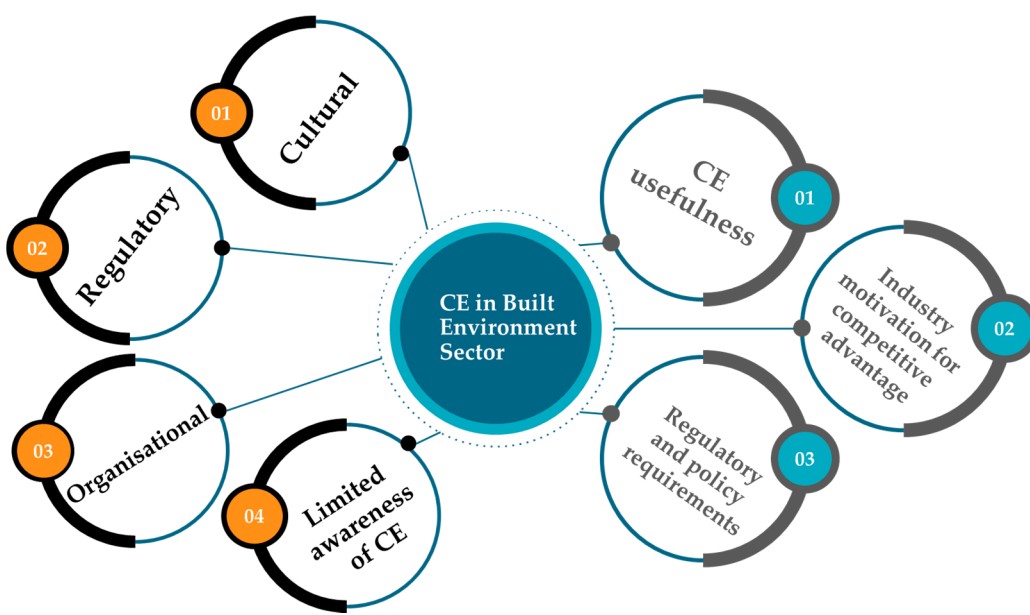

**Figure 6.** Categories of barriers and enablers of CE application in the built environment sector.

3.2.1. Cultural Barriers

There are some cultural aspects and social norms that heavily influence the building industry. First, the aesthetic preferences of designers, developers, and building occupants

may be incongruent with some of the strategies to enable circular economies in the building sector [41]. Traditionally, embellishment of buildings shows wealth and/or status, particularly in residential settings. Secondly, another cultural barrier is the high-consumption lifestyles that have spread in many cities around the world. Lastly, professionals in the building sector tend to be resistant or slow to change and wary of incorporating new approaches [41].

### 3.2.2. Regulatory and Policy Barriers

Legislation gives little attention to design phases which are essential to waste minimisation [37,42]. In Australia, there is a lack of supportive policies, regulations, and standards to foster circularity in the built environment [43]. Several issues impede the implementation of these essential regulatory measures. There are issues with the reclassification of waste as a resource, which hampers EoL treatment and material reuse. Additionally, there are inconsistencies in waste and materials management regulations across Australia, making it difficult to produce national frameworks [43].

### 3.2.3. Organisational Barriers

In Australia, some barriers at the organisation level inhibit the progress towards a more circular built environment, as was evidenced in a recent Green Building Council of Australia (GBCA) survey [43]. Demands such as emissions reduction or carbon footprint reductions take precedence. Likewise, the survey revealed that there is not enough stakeholder collaboration and a lack of agreed metrics and processing infrastructure.

### 3.2.4. Limited Awareness and Understanding of CE

In Australia, there is little understanding of what a CE can represent to the building sector and the benefits it provides [37,43]. In addition, there is a need to differentiate between a CE approach and a traditional sustainable model. The former is based on strategies that make business sense, while the latter has moved away from financial aspects. This can certainly discourage the uptake of a CE [44]. Overall, if there is no understanding of the value of a CE, there will not be a real drive to push the building industry into circularity [44]. A survey conducted on several industry stakeholders in Australia revealed that participants' awareness of the CE has a statistically significant impact on aspects including willingness to adopt a CE, CE adoptions, and benefits of the CE [37].

### *3.3. Key Enablers*

From the selected data and meta-analysis, the following key enablers of the application of a CE in the built environment were identified (Figure 6). The three top enabling clusters are the aspiration for sustainability, competitive advantage/cost reduction, and regulatory requirements. The literature examined in this section covers both Australian and other countries' sources.

### 3.3.1. CE Usefulness

Research has shown that there are several driving factors related to sustainability that can enable a CE in the building sector. These include resource and energy conservation, enhancing indoor environmental quality, and waste minimisation [45]. Providing evidence of CE-added value has been identified as a major driver to implementing CE in the Australian built environment [37].

### 3.3.2. Industry Motivation for Competitive Advantage/Cost Reduction

Investing in sustainable solutions may be seen as a cost premium in a project building, and there could be long payback periods before recovering the initial investment [41]. However, provided that 50–60% of construction project costs account for materials, reducing waste is a cost-saving strategy [46]. This means that material waste heavily contributes to cost overrun. For instance, it has been shown that waste materials such as concrete, piles,

and stone tablets increase project overrun by 7%, 13%, and 26%, respectively [46]. In short, not only do material cost savings improve financial performance, but they also can enable circularity in the built sector.

### 3.3.3. Regulatory and Policy Requirements

The policy is essential to enable a CE in Australia's building sector. Stringent legislation and policy have been identified as key success factors in enabling DfD of buildings, which lends itself to designing out waste [42,47]. Waste construction effectiveness is dependent on the extent to which it was designed out [42]. Governments must set building deconstruction targets and provide incentives to achieve them to reduce waste intensiveness [48]. This is a proven way to enable DfD of buildings as construction works require planning acceptance and authorisation based on current regulatory frameworks [47]. Other policies that have been identified as (potential) waste minimisation enablers in the construction industry are [42]:

○ Economic carrots: these could be in the form of tax reliefs or incentives. These measures are especially welcomed by construction contractors and site workers, who are highly motivated by the economic benefits of waste minimisation [49]. Construction industry experts and workers would be incentivised (not coerced) to reduce waste. A study conducted in Australia showed that relevant stakeholders perceive the lack of incentives as an important obstacle to CE implementation in the building sector [37].
○ Sustainable construction appraisal systems: most of these tools allocate points for material sorting, recycling, and/or reusing (e.g., US's LEEDS, UK's BREAAM). The same point allocation mechanism could be applied to waste management, which could be a key factor in designing out waste.

### 3.4. SWOT Analysis

The data collected from the selected literature were analysed to identify strengths, weaknesses, opportunities, and threats (SWOT). The SWOT analysis (Table 5) provides an initial understanding of all data collected.

**Table 5.** SWOT matrix developed to gain an initial overview of selected data.

| Strengths | Weaknesses |
|---|---|
| Circularity in the built environment is gaining traction, as there are more documents studying this matter every year. There is consistency among information from different sources; there are no contradictory recommendations and/or strategies as to how to achieve circularity in the built environment through DoW. | Theoretical methods are more popular than case studies or application approaches. Many studies focused on the positive outcomes for the environment when applying circular methods; economic and social aspects are often disregarded. |
| **Opportunities** | **Threats** |
| Few studies/documents mention the social contributions of promoting circular built environments. Design and EoL life-cycle stages seem to have more circular strategies than other phases, namely, operation. | There is hardly any information that supports or proves to the industry why applying circular practices is worthwhile, especially economic-wise. Most cases of success have been trialled in other countries, with different policies and contexts to Australia. |

## 4. Circular Economy Guidelines for the Construction and Infrastructure Industry

Analysis of both scholarly and grey literature led to the development of guidelines for implementing CE practices in the construction and infrastructure industry. The guidelines were based on the broader scopes identified during the literature review. Table 6 shows an overview of the developed guidelines. Performance sustainability measurements in

the form of either qualitative or quantitative key performance indicators (KPIs) are provided for the strategies shown in Table 6. In addition, each strategy was evaluated using a heatmap according to the level of alignment with the Ellen McArthur Foundation (EMF) CE principles. The levels of alignment are high, medium, and low. A high alignment means that the strategy could contribute greatly to achieving the desired outcomes of each EMF CE principle, a medium alignment means a contribution to a certain extent, and a low level means a low contribution. The questions below were considered for sorting the strategies into each of these levels.

(i) 1st EMF CE principle: DoW and pollution
Does the strategy design out the negative human and environmental impacts of the economic activity by avoiding the release of GHG and hazardous substances? Does it avoid the release of pollutants into the air, land, or water? Does it avoid the generation of waste, including structural waste?

(ii) 2nd EMF CE principle: keep products and materials in use
Does the strategy help preserve the value of materials by designing for reuse, longevity, and recyclability? Does the strategy include designs for adaptability or flexibility or any other actions that keep products and materials circulating in the economy?

(iii) 3rd EMF CE principle: regenerate natural systems
Does the strategy actively improve the environment? Is the use of non-renewable resources avoided? Is there any improvement in solid, air, or water? Does the strategy support the use of renewable energy?

**Table 6.** CE Guidelines for the built environment and alignment with the EMF CE principles.

| Broader Scope | Strategy | KPI | Alignment with EMF CE Principles | | |
| --- | --- | --- | --- | --- | --- |
| | | | DoW and Pollution | Keep Products and Materials in Use | Regenerate Natural Systems |
| Circularity practices | 1. CE procurement | Both qualitative (yes/no) and quantitative (%) of the PwRC | High | High | Medium |
| | 2. Design for Deconstruction (DfD) | Disassembly Potential Rating | High | Medium | Low |
| | 3. Design for flexibility and adaptability | Adaptability and Flexibility Rating. Intensity use = proportion of the building's UA/GFA | Medium | High | Low |
| | 4. Design for long life | Both qualitative (yes/no) and quantitative whole LCC [$/m$^2$/year] | High | High | Low |
| | 5. Eliminate building components | Material use intensity per functional unit (kg/unit/year) | High | Low | Low |
| | 6. Reuse building/building elements | Reused floor area (% of total GFA) or building component reuse in (%) | High | High | Medium |
| | 7. Restore and regenerate | Soil sealing factor and/or compensatory measures (rainwater management, green roofs) | Low | Low | High |
| | 8. Design out hazardous/pollutant materials | Environmental Impact Cost [$/m$^2$/year] | Medium | Low | High |
| | 9. Climate resilient design | Embodied Carbon Intensity [kgCO$_2$ eq/m$^2$/year] | Medium | Low | High |
| | 10. Sharing economy/shared space | Provision of the shared economy (yes/no) or % of shared space | Medium | High | Low |
| Resource management | 11. Waste prevention on the construction site | Diversion rate from landfill (%) | Medium | High | Low |
| | 12. Material/component recycling | Proportion (mass/unit of reference) of secondary materials installed in the building. GFA could serve as the unit of reference | High | High | Low |

**Table 6.** *Cont.*

| Broader Scope | Strategy | KPI | Alignment with EMF CE Principles | | |
| --- | --- | --- | --- | --- | --- |
| | | | DoW and Pollution | Keep Products and Materials in Use | Regenerate Natural Systems |
| Innovation and optimisation | 13. Use of digital technology (e.g., material passports) | The proportion of building components or traceable materials in % | Medium | Medium | Low |
| | 14. Construction Innovation (e.g., modular construction) | Proportion modular and/or off-site construction in % | Medium | Low | Medium |
| | 15. Green supply chain (e.g., use of bio-based materials) | Local vs overseas/sustainably sourced (%) | High | Low | High |
| | 16. Use and integration of sustainable technology | Energy demand from renewable sources (%), Energy storage capacity (kWh/time), microgrid options, etc. | High | Low | Medium |

*4.1. Discussion*

The sections below describe the strategies in each of the three broader scopes identified in the literature review: circularity practices, resource management, and innovation and optimisation.

4.1.1. Circularity Practices

- Circular economy procurement

A number of strategies have been developed to include sustainability aspects in the tender process. By integrating sustainability right from the tender phase, the decisions can take an integrated, holistic approach and no longer have to rely solely on economic factors [50]. Explicit tender requirements that target the impact on health and the environment would be integrated into the invitation to tender. Technical requirements, such as durability, ease of recycling, or ease of recovery, can also be included in this phase. Certification organisations such as the German Sustainable Building Council (DGNB) award tender requirements that explicitly request or recommend the use of secondary materials or the reuse of materials for mineral construction products. Toxic and hazardous materials must not be accepted during the procurement stage. Other targets identified may come as PwRC in the tender and/or construction stages.

Procurement can also be decisive in controlling waste by improving logistics [51]. Waste-efficient procurement actions, including just-in-time delivery, improved supply chain collaboration, and reduced packaging [42,52]. These actions can avoid excessive packaging materials, inadequate materials storage, and double or poor handling, all of which are associated with construction waste [42]. Similarly, another successful action that can be taken during the procurement process is to include a contractual clause in which sub-contractors are held accountable for the disposal of their waste [48,53]. A study found that in the Australian context, construction project managers consider that incorporating waste management in the design phase is more essential than effective on-site waste management [54].

- Design for Deconstruction (DfD)

Deconstruction is the careful, piece-by-piece disassembly of buildings [47,55]. Deconstruction of a building is also known as selective demolition or disassembly [55]. The main goal is to maximise the potential reuse and recovery of a building's components and materials [47,55] and to prevent demolition at the EoL [47]. The benefits of conducting deconstruction processes outweigh the cost so long as the value of the building component is preserved when it reaches its end-of-life [47]. In fact, through DfD protocols, the value of a building can be retained as well as increased [10]. If buildings are designed for disassembly, a wide variety of end-of-life scenarios for the building's components and materials

can be reached, such as reusing, recycling, relocation, and composting [56]. Table 7 shows actions that have been identified as key when conducting DfD.

**Table 7.** Actions that support DfD.

| Actions | Key Reference/s |
|---|---|
| Allow parallel rather than sequential disassembly. | [56,57] |
| Use lightweight materials to facilitate the easy handling of components. | [47,55,56] |
| Size components to suit the proposed means of handling. | [56] |
| Separate structure from cladding to allow changes to the building envelope. | [56] |
| Provide access to all parts of the building that are to be disassembled | [56] |
| Arrange components in a hierarchy of access related to life spans. | [10,56,57] |
| Use a modular system that is compatible with existing standards. | [55,56] |
| Reduce, simplify, and standardise connections. | [10,56] |
| Provide a means of identification of components and assembly instructions. | [56,57] |
| Design using an open system that allows for structural alternatives. | [47,56] |
| Allow for disassembly at all scales, from materials to whole buildings. | [56] |
| Logistics/manual of disassembly. | [56] |
| Avoid cast-in-place composite systems unless they are recyclable and reusable and do not cause negative environmental impacts. | [47,57,58] |
| Avoid the use of joints and/or screws. | [47,55,56] |
| Avoid the use of chemical connections (e.g., adhesives, coatings). | [10,47,55] |
| Avoid the use of hazardous materials and compounds. | [10,47,55,57] |

- Design for flexibility and adaptability

  This design seeks to provide the building with multiple life cycles to make the most of resources and materials in terms of spatial and technical domains [59]. The former entails the ability to transform with low resource consumption (if any), and the latter involves structural and material alterations to reuse materials in the future [59]. Some specific actions include:

  ○ Increase convertibility: Allow for changes in building use by designing the building envelope to allow for more than one use or to allow modifications in window size and spacing [59];
  ○ Use standard, simple construction tools and technologies [55–57];
  ○ Avoid bespoke/tailor-made solutions and complex building geometries [60].

- Design for Longevity

  A key objective is to keep the value of materials and resources as long as possible by intensifying their use [59]. The means to realise this is by extending the service life of buildings by (i) specifying durable components [41], (ii) avoiding the use of synthetic materials that do not allow refurbishment [41], (iii) prioritising standardised, modular elements, (iv) maximising the durability of the building structure through careful selection, protection, and maintenance of components, (v) making use of Whole Life-Cycle Cost assessment (WLCC) as a design assessment tool, (vi) assembling in a systemic manner that is suitable for maintenance and allows for the possibility of replacements, and (vii) designing for longer service life [41].

- Eliminate building/building elements

  This strategy is underpinned by dematerialisation principles, which seek to reduce materials or resource inputs in buildings while increasing performance and functional-

ity [41]. These principles have many similarities with the concepts of eco-efficiency and material/resource efficiency that have been researched and applied in the construction sector. The concepts aim for economically competitive products that result from using materials and resource inputs efficiently. Dematerialisation considers all the environmental impacts (direct or indirect) of the life cycle of a product from the design stage. Several dematerialising strategies can be applied to a building project [41], as illustrated in Figure 7.

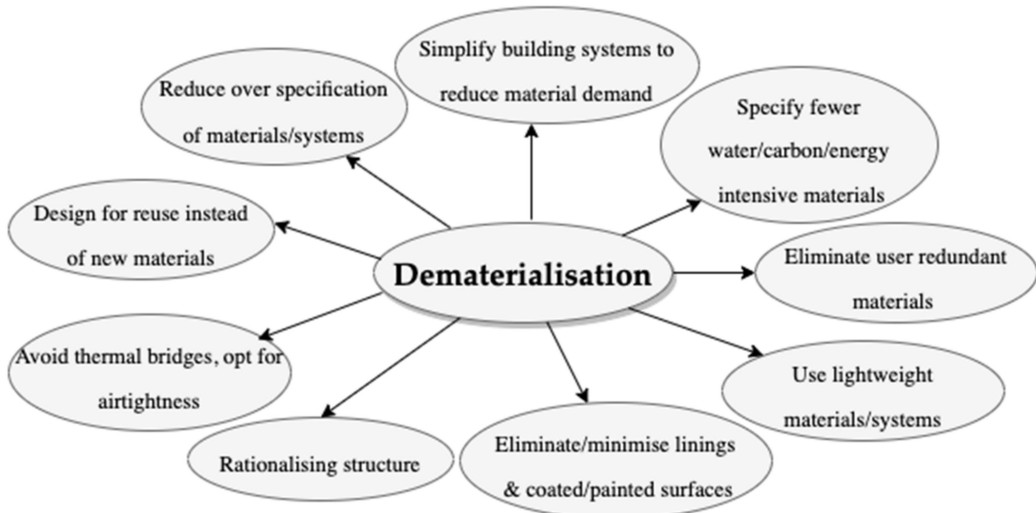

**Figure 7.** Summary of dematerialisation principles. Source: Authors.

- Reuse building/building elements

According to the waste management hierarchy model [61], reusing waste resources is superior to recycling due to its environmental benefits, such as reduction in energy consumption and GHG emissions, the two major criticism of using PwRC in the sector [62]. Furthermore, the technical advantage of reusing building elements is that they are reintroduced into the economy without requiring significant structural modifications or additional resources. Reuse can be performed either in the same or a new building location, and the function of the material or product may be different. Some examples of reusing building components include the reuse of greywater [63] and the adaptive reuse of existing structures [41]. Even though there are hurdles to overcome to reuse buildings or building components, which can be social, economic, material, or stakeholder related, the main obstacle is that they have not been designed for this purpose [56]. Therefore, DoW needs to be in alignment with sustainable manufacturing objectives.

- Restore and regenerate

This strategy aims to generate a positive impact on human and natural systems. Regenerative buildings repair, recreate, and revitalise their sources of water, energy, air, waste, or any other matter [64]. Some of the actions include:

○ Accommodating green spaces in and out of buildings to promote biodiversity and interaction between people and the surrounding nature. For example, by implementing green roofs, urban agriculture [59], trees, parks, gardens, green facades, or permaculture [64].
○ Allowing high-quality indoor and outdoor environments to improve air quality. This could be achieved by implementing naturally lit/ventilated spaces or using vegetated walls or suspended gardens [64].
○ Incorporating eco-friendly in-situ water treatment systems for both blackwater and greywater. The systems can be part of the greenery outlook and improve biodiversity [64].

○　Soil sealing: Sparing use of land that minimises the impact on this land at a local level results in lower development, wastewater charges, and an improved microclimate [65].

○　Implementing brownfield redevelopment processes when there is contaminated land to make it suitable for reuse. The existing land is significantly improved if the soil on the plot is properly disposed of. Soil can be deemed as contaminated not only when it is already classed as such but also if there is suspicion of contamination (polluting activities, spills, accidents) or if it contains munitions [66].

- Design out hazardous/pollutant materials

Manufacturers must assess their materials according to the hazards of materials as well as the exposure routes during the use and EoL stages [64]. Materials that may be exposed to humans and nature should not contain harmful chemicals. If hazardous products are needed for any reason, they should be identified with a tag so they can be correctly handled at the end of their use [67].

- Climate resilient design

The main goal of this strategy is to minimise the embodied carbon footprint of materials. The actions to realise this involve two main approaches:

-　Using eco-friendly alternative materials: (a) bio-based products and finishing materials which can significantly reduce the embodied carbon of a building [64] (e.g., sustainably sourced timber by cement); (b) reused or recycled materials, preferably locally sourced (fewer transport emissions) (e.g., reclaimed bricks, locally recycled aggregates); (c) high-durability, low-maintenance materials (e.g., components with same lifespans as the building).

-　Using fewer materials: (a) design optimisation leads to a reduction in quantities of materials used [10] (e.g., compact buildings that enable natural ventilation, lower wall/floor ratio); (b) design for deconstruction; (c) design for less on-site waste (e.g., off-site construction).

A good practice that can be implemented towards the achievement of this strategy is tracking and tracing the embodied carbon footprint of the building parts (e.g., structure, envelope, systems, fittings) and specifying a target below the recommended limits of the local area (ARUP). The use of digital technologies can facilitate this practice.

- Sharing economy/shared space

This strategy is concerned with intensifying the functional use of materials. This may involve a reduction in floor area, incorporation of shared economy/functions, and designing for multi-functionality and adaptability [41,59]. It must be stressed that none of these actions should hinder the building's capacity to meet basic user needs and/or operational performance [41].

4.1.2. Resource Management

- Waste prevention on the construction site

Innovative processes and concepts have paved the way for waste reduction at the construction site. Off-site construction considers the prefabrication of components, precasting of structural elements, and design for off-site construction [42]. By constructing off-site, waste-generating factors such as unexpected design changes, materials handling, and poor storage are completely prevented [42]. Indeed, ensuring a few design changes during the construction process has been identified as one of the top strategies for on-site waste minimisation. Provided that off-site construction helps reduce the number of design changes, this is an effective strategy to minimise on-site waste [48]. Other key strategies to combat waste in the construction site are:

○　Following the project drawing/designs as closely as possible [48];
○　Identifying potential problems to improve the quality of the final design [68];

○ Assigning a dedicated place to collect and sort waste [48];
○ Setting up waste bins in each building zone of the construction site [48];
○ Optimising material waste segregation by providing skip bins for specific materials [48];
○ Providing designers with technical information and capabilities of materials and equipment [68];
○ Instigating synergistic interactions with other industries to recover the value of construction waste and by-products (e.g., Industrial Symbiosis where the waste of an industry is used as a resource in another industry) [69].

- Material/component recycling

The use of secondary products may not necessarily contribute to ongoing project waste reduction, so contractors do not see it as an effective strategy to reduce construction waste. However, using PwRC is both essential to increasing waste diversion from landfills and decreasing waste intensiveness in the construction sector [48]. Using secondary materials also reduces the demand for raw materials extraction and their associated impacts [70]. Thus, PwRC should be used in construction projects. A self-declaration or manufacturer declaration specifying the material, component, or PwRC must be requested. Secondary raw materials contents can be either manufacturer-specific or sector-specific. Recycling approaches can also be considered when using aggregates in components (e.g., recycled concrete aggregate) or when deciding on fixtures or fittings [41]. It is worth mentioning that recycling should only be practised if it saves energy and emissions compared to primary production and that it is technically feasible. Having said that, sometimes recycling leads to loss of quality, and sometimes there are no recycling routes for materials that retain their initial structure and quality (e.g., cement) [71].

### 4.1.3. Innovation and Optimisation

- Use of digital technology

Although the built environment has not seen the same digital technology (DT) progress as other sectors, there have been important developments in the last decades [59]. These days, there are a variety of technologies that can be used in different life cycle stages of a building. Some of the DTs include Artificial Intelligence (AI), BlockChain Technology (BCT), Building Information Modelling (BIM), Digital Twins (DTwins), and Material Passports and Databanks, among others. DTs can bring about several benefits to the building sector. Firstly, they foster supply chain collaboration among stakeholders as they can promote value networks, data-sharing, and knowledge. This is of the utmost importance as collaboration has been identified as a key factor in facilitating the CE in the building sector [59]. Secondly, they reduce the demand for raw materials by streamlining design processes, thus reducing waste generation. Thirdly, DTs help track and trace materials and resources throughout the entire lifespan of a building, which permits them to capture embodied value when they reach their end-of-life [59]. Overall, DTs help to virtually represent a built asset with all relevant building information, including material properties, maintenance schedules, (dis)assembly, and operation manuals. With DTs, buildings are seen as material banks, which creates new end-of-life possibilities for these materials other than becoming waste.

- Construction Innovation

Several methods that are different from traditional in-situ construction activities can be implemented to support a CE in the building sector. Non-conventional construction methods include standardised and modular designs, as well as the repetition of similar designs in more than one component. Both methods rank as the most popular prefabrication strategies. Some of the benefits of prefabrication include higher quality of the final product, reduction of material demand and waste generation, and improved health and safety [72].

- Green supply chain

In terms of bio-based materials, a decisive factor in selecting construction materials is quality-cost effectiveness. Organic-based or ecofriendly (sustainably sourced) materials are often overlooked as they are more labour-intensive and/or more expensive than traditional construction materials [73]. A study found that the replacement of traditional materials in large amounts by more eco-friendly ones can be considerably challenging, as the quality of the traditional materials is already well established, and replacement could be an issue factoring into sector and market trends [73]. Policies that incentivise the application of more sustainable materials are the key to changing the situation. More companies would be pushed to use greener construction materials [73]. Along with policies that drive the selection of sustainably sourced materials, there are some requirements that construction companies can demand from their materials suppliers:

○　A raw materials list documenting origin, extraction, processing stages, and locations where the processing takes place.
○　A corporate mission statement that supports the prevention of adverse environmental impacts from their raw materials production process activities. Other social corporate statements, including the prevention of human rights abuses and corruption, should also be requested.

- Use and integration of sustainable technology

This strategy plans to capture economic value from sustainable and regenerative building technologies [59]. Specific actions include:

○　Using clean/renewable sources of energy which should be locally generated [66];
○　Applying Passive House or Minergie designs to ensure minimum energy consumption and maximum thermal comfort [64];
○　Grid compatibility: Energy that is generated in the building (positive energy generation) or on its land from renewable energy sources is fed to the district/the area in the immediate vicinity [66]. Positive impact buildings have technologies that enable such transfer [59].

## 5. Conclusions

This study set out to develop CE guidelines for the C&D waste management of the Australian built environment sector. Although there are various design methods to minimise C&D waste, DoW guidelines have yet to be clearly outlined. This study sets a framework for the successful implementation of these guidelines. In doing so, a descriptive literature review was conducted to understand the current global and Australian CE practices in the sector. A main contribution to the theoretical knowledge is the identification of a significant research gap in terms of circular strategies during the operational stage of the building, as most articles focused on the EoL. Furthermore, as a practical contribution, the research provides multiple industry-driven strategies that help the built environment sector to achieve CE objectives for improved circularity in the resources in use.

This study also identified the main clusters of barriers and enablers of CE implementation in the built environment sector. Analysing these factors will help decision-makers to make informed decisions about the extent to which CE principles can be applied in the sector. Finally, this paper outlined industry-specific guidelines for implementing CE practices. Below are the main takeaways from these guidelines that will have sustainability management implications in the built environment sector:

- The guidelines present 16 strategies with detailed actions that are applicable to all stages in the value chain, from design, manufacture, and operation, to EoL. The project managers in the sector can adapt and apply these strategies where possible.
- The strategies encourage the sector to move away from the traditional, linear way of managing the built environment, not only by keeping construction materials in the loop and by improving waste management performance but also by using the built

environment to regenerate nature (e.g., climate-resilient design). The latter is particularly important in delivering future projects where the demonstration of achieving (carbon) emission reductions through climate-resilient design is necessary.

- The seven categories identified as having a literature-specific focus (CE/Zero waste, Construction Innovation, DfD/Circular design, Digital Technology, Project/Supply Optimisation, Resource Management, Reuse, and Sustainability Assessment) served as a foundation to develop the three main scope areas of the proposed guidelines (circularity practices, resource management, and innovation and optimisation). Paying attention to these scope areas is necessary from the project management perspective to deal with barriers they face throughout the process of transforming to circularity.
- The circularity practices have a strong focus on various design concepts, namely, DfD, design for flexibility and adaptability, design for long life, design out hazardous/pollutant materials, and climate-resilient design. Resource management strategies are relevant when the material reaches its EoL. Innovation and optimisation strategies look to make the most of technological breakthroughs to realise circular built environments. Hence, designers and project managers in the sector need continuous education and training on the latest technological advancements to ensure that circularity will be business as usual.
- The literature review uncovered targeted strategies and actions that contribute to the DoW and pollution and keeping resources in use (two of the EMF CE principles)

Most notably, the CE guidelines for the built environment suggested in this study should be seen as a dynamic framework that should be updated and revised with new research, good practices, technologies, and continuous learning.

*Further Research*

From the literature review and the meta-analysis, the following research topics and questions emerged:

- Although the guidelines include strategies that help regenerate natural systems, the third EMF CE principle, there are still far more strategies that cover the other EMF CE principles. What else could be done to actively improve the natural environment?
- Another possible area for further research is to assess the Scope 3 savings linked to the strategies of these guidelines. This includes assessing the actual energy savings of using PwRC in construction. Will the energy spent to incorporate waste in new construction projects be less than the current energy demand?
- Given the non-negligible impact of social factors in establishing the CE in the sector, it is vital to understand the approach to assessing and/or determining the social relevance of the guidelines and, more specifically, designing out C&D waste in Australia. There is a clear lack of information regarding social aspects and circularity in the built environment.
- Research that compiles and assesses techniques to include waste in construction through case studies or lessons learned could be of great use to the industry.

**Author Contributions:** Conceptualisation, A.Z., T.M., S.S., T.R. and S.C.; methodology, A.Z., S.S. and T.M.; formal analysis, A.M.C.R. and A.Z.; writing—original draft preparation, A.M.C.R. and A.Z.; writing—review and editing, A.M.C.R., S.S., T.R., S.C. and A.Z.; supervision, A.Z. and T.M. All authors have read and agreed to the published version of the manuscript.

**Funding:** This research has been developed with support provided by Australia's Sustainable Built Environment National Research Centre (SBEnrc). SBEnrc develops projects informed by industry partner needs, secures national funding, project manages collaborative research, and oversees research into practice initiatives. Core Members of SBEnrc include ATCO Australia, BGC Australia, Government of Western Australia, Queensland Government, Curtin University, Griffith University, RMIT University, and Western Sydney University. This research would not have been possible without the valuable support of our core industry, government, and research partners.

**Institutional Review Board Statement:** Not applicable.

**Informed Consent Statement:** Not applicable.

**Data Availability Statement:** The data presented in this study are available on request from the corresponding author.

**Conflicts of Interest:** The authors declare no conflict of interest.

## Abbreviations

| | |
|---|---|
| ACT | Australian Capital Territory |
| AI | Artificial Intelligence |
| BCT | Blockchain Technology |
| BIM | Building Information Modelling |
| C&D | Construction and Demolition |
| CE | Circular Economy |
| DfD | Design for Disassembly |
| DfR | Design for Recycling |
| DfZW | Design for Zero Waste |
| DoW | Design out Waste |
| DT | Digital Technology |
| EMF | Ellen McArthur Foundation |
| EoL | End-of-life |
| GFA | Gross Floor Area |
| GHG | greenhouse gas |
| KPI | Key Performance Indicator |
| LCA | Life-Cycle Assessment |
| Mt | Million Metric Tonnes |
| NSW | New South Wales |
| NT | Northern Territory |
| PwRC | Products with Recycled Content |
| QLD | Queensland |
| SA | South Australia |
| Tas | Tasmania |
| UA | Usable Area |
| Vic | Victoria |
| WA | Western Australia |
| WLLC | Whole Life-Cycle Cost |
| WoS | Web of Science. |

# Appendix A

**Figure A1.** *Cont.*

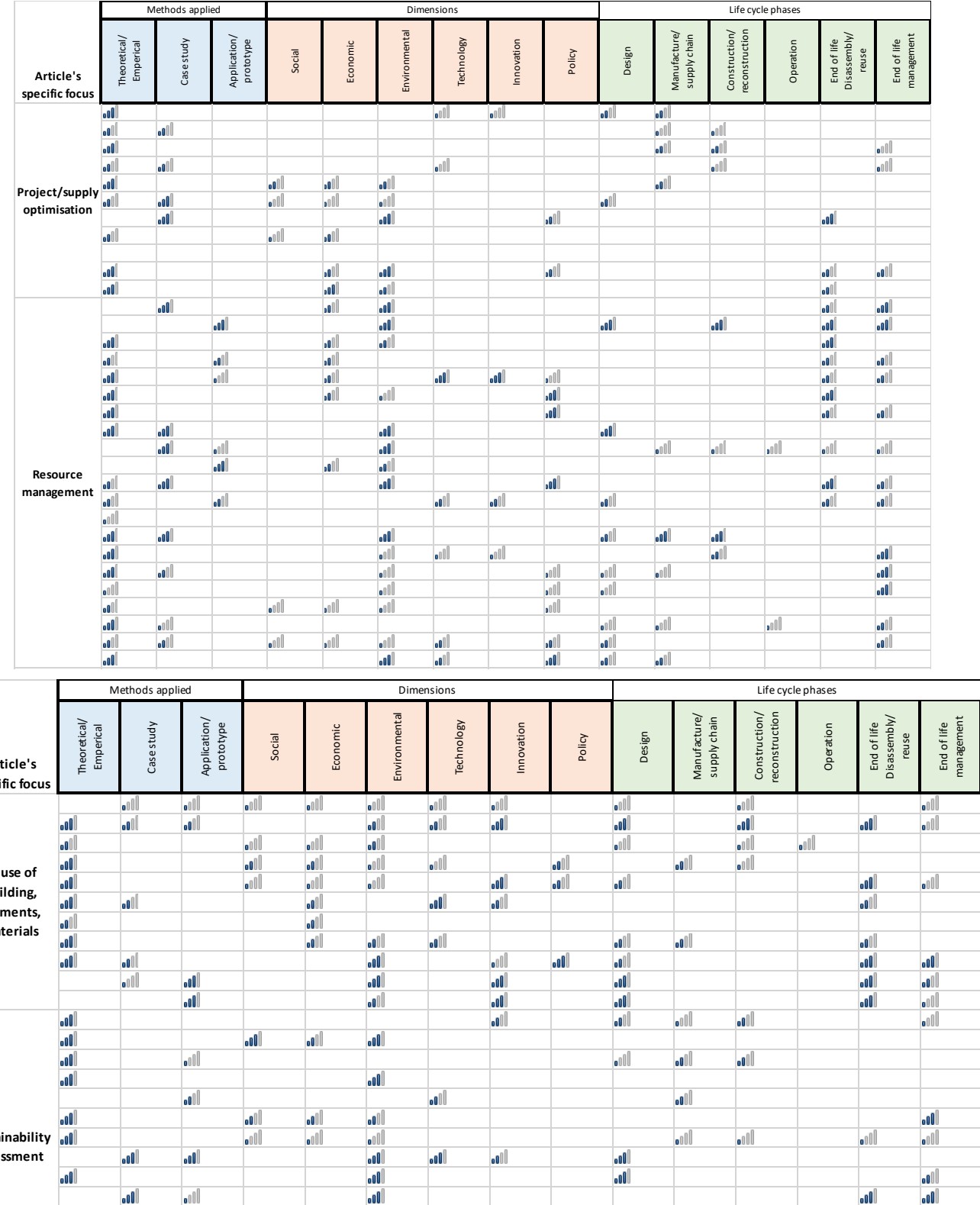

**Figure A1.** Meta-analysis on selected articles.

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
