# Peer review of "Development of the Circular Economy Design Guidelines for the Australian Built Environment Sector"

_sustainability, doi:10.3390/su15032500_

Round 1

Reviewer 1 Report

The paper is comprehensive and well-organized. Some issues should be considered in the revised manuscript.

- In the abstract section, the significant outcomes with respect to the main challenges should be mentioned.

- In the conclusions, The novelty of the research and its practical usefulness should be indicated. Indicate the limitations of the research and to what extent the study fills a gap in the literature.

- Section 3.2. seems messy. The selection of the barriers and challenges is not explained—likewise section 3.3. A brief introduction with explanations is needed.

- The literature review process of this article is quite clear. However, suggestions for management implications can be strengthened in conclusions.

- In conclusion, the social relevance of this work is not highlighted as well.

Reviewer 2 Report

Dear Authors,

Congratulations on your work, which is focused on a very interesting subject. As any other paper in this phase, there are some amendments to do, whose can improve the overall quality of your paper. Thus, I'm providing below some comments and suggestions, trying to collaborate by this way in improving your paper:

1. The purpose of your work is very interesting, but is too much focused on the Australian reality. There are a lot  of techniques developed in the last years to integrate a lot of construction waste in new buildings. This is only lightly discussed in your paper. 

2. Regarding the Conclusions, I'm not sure that your study can be generalized for other industries or other continents in the same way. Please try to highlight your main contribuions to the theoretical/academic knowledge.

3. You refer that End-of-Life is one of the most considered ways regarding civil construction and demolitions. Did you have read the book: Cleaner Production - Toward a Better Future, published by Springer in 2020? Maybe you get some important information there (Pages 201, 335, 395).

4. Construction is responsible for about 30% of the energy consumption worlwide. Did you have care with the energy spent to integrate waste in new construction?

5. I have dealt with the integration of fiber glass reinforced plastics in mortars, but I'm aware that the flow of production of these products and others are higher than the reuse that is possible to carry out technically. Please comment this.

6. Construction is going to robotized means, even using Additive Manufacturing technologies. How your work can deal with this new approach? Please comment.

 7. Your work remains extremelly focused on Australia reality, namely referring the goals for 2030. However, you know that regulations in other countries, not rich such as Australia, are very different. Thus, could your framework be applied to low-income countries, or under-developed countries? We know that when nothing exists, it is easy to built anything. However, the deorganization of some countries in Asia and Africa present a really different reality and need a different approach. Please comment.

8. You are describing the three priorities of the Australia Northern Territory Government. However, dou you believe on them? We are seeing COP 27 in Sharm El-Sheikk, without more promises, but forgotting the previous agreements and almost nothing doing to reverse the environmental degradation. Thus, these objective really matter?

9. In your Methodology, please define the range of time used for search (from 2xxx to 2xxx), the databases consulted (just SCOPUS and WoS?), and the criteria to include or not different papers (journal not indexed, and so on).

10. How did you selected the non-relevant papers?

11. In Figure 4, the title of the graph is redundant with the figure caption.

12. Before Chapter 4, a deep SWOT analysis would be welcome, in order to highlight what was found as really important, what is secondary, ways of research and so on.

13. In 4.1.1., there is a real asymmetry between the literature support achieved for Circular Economy and DfD. Could you improve the first?

14. There also are other topics with a low previous support.

15. The conclusions should be structured in a way that are able to help the reader to find the most important findings brought by your work. Using topics/bullet points, I think the reader would be grateful.

Hope have helped in improving your paper.

Kind regards.

Reviewer 3 Report

This paper studies an important topic the circular economy. The authors conducted a meta-analysis to explore the current CE practices in construction and infrastructure projects in the global and Australian context. I think this paper has great potential to grow into a publication. Here are my concerns for this paper.

1. The paper does not address the file drawer problem. The file drawer problem rests on the assumption that statistically non-significant results are less likely to be published in primary-level studies and less likely to be included in meta-analytic reviews, thereby resulting in upwardly biased meta-analytically derived effect sizes. The authors need to address this issue.

2. The paper lacks effect size analysis. The nature of the meta-analysis in this paper is descriptive. The authors need to provide more evidence about the key findings in the extant literature using statistical analysis (e.g., effect size analysis). T

3. The high/medium/low levels are not defined in Table 5. The authors used high/medium/low levels to classify the extent to which "Alignment with EMF CE principles." However, the authors did not give any definition about these levels. The authors need to provide that. 

Good luck with this interesting research.

Round 2

Reviewer 1 Report

The authors have addressed the comments. The paper can be accepted from my side.

Reviewer 3 Report

I recommend accepting this paper.